# Surface Characteristics and Color Stability of Dental PEEK Related to Water Saturation and Thermal Cycling

**DOI:** 10.3390/polym14112144

**Published:** 2022-05-25

**Authors:** Liliana Porojan, Flavia Roxana Toma, Mihaela Ionela Bîrdeanu, Roxana Diana Vasiliu, Ion-Dragoș Uțu, Anamaria Matichescu

**Affiliations:** 1Department of Dental Prostheses Technology (Dental Technology), Center for Advanced Technologies in Dental Prosthodontics, Faculty of Dental Medicine, Victor Babeș University of Medicine and Pharmacy Timișoara, Eftimie Murgu Square No. 2, 300041 Timișoara, Romania; flavia.toma@umft.ro (F.R.T.); roxana.vasiliu@umft.ro (R.D.V.); 2National Institute for Research and Development in Electrochemistry and Condensed Matter, 300569 Timisoara, Romania; mihaelabirdeanu@gmail.com; 3Department of Materials and Fabrication Engineering, Politehnica University Timişoara, Bulevard MihaiViteazul nr.1, 300222 Timişoara, Romania; dragos.utu@upt.ro; 4Department of Preventive, Community Dentistry and Oral Health, Center for Advanced Technologies in Dental Prosthodontics, Faculty of Dental Medicine, Victor Babes University of Medicine and Pharmacy Timișoara, Eftimie Murgu Square No. 2, 300041 Timișoara, Romania; matichescu.anamaria@umft.ro

**Keywords:** dental PEEK, surface characteristics, microhardness, color changes, water saturation, thermal cycling

## Abstract

(1) Background: The study was undertaken to evaluate the surface characteristics, microhardness, and color stability of PEEK materials related to water saturation and in vitro aging. (2) Methods: Custom specimens of unmodified and modified PEEK CAD/CAM materials were investigated: BioHPP, a ceramic reinforced PEEK, and Finoframe PEEK and Juvora medical PEEK, 100% PEEK materials. Forty-eight plates were sectioned in rectangular slices. The specimens were immersed in distilled water at 37 °C for a period of 28 days, and then subjected to aging by thermal cycling (10,000 cycles). Surface roughness was measured with a contact profilometer; nanosurface topographic characterization was made by Atomic Force Microscopy; Vickers hardness measurements were performed with a micro-hardness tester; color changes were calculated. All registrations were made before immersion in water and then subsequently once a week for 4 weeks, and after thermocycling. (3) Results: The studied reinforced and unfilled PEEK materials reached water saturation after the first week of immersion, without significant differences between them. The most affected from this point of view was the reinforced PEEK material. Thermocycling induces a significant increase inmicroroughness, without significant differences between the studied materials. In relation to the nanosurface topography and roughness, the reinforced PEEK material was the least modified by aging. The color changes after 4 weeks of water immersion and one year of simulated in vitro aging ranged from extremely slight to slight, for all materials. (4) Conclusions: Water absorption was associated with a decrease in microhardness. Surface characteristics are affected by water immersion and thermocycling. Perceivable or marked color changes of the materials were not detected during the study.

## 1. Introduction

Advances in removable partial denture technology are strongly related to the development of dental materials. The most commonly used materials for removable partial dentures frameworks are metal alloys, allowing producing precise structures with a high strength and adequate retention and friction, but an important disadvantage is the corrosion instability [1].

Due to the direct contact with large areas of the oral mucosa, there is a potential for topical disorders and allergic reactions. The aesthetics of conventional removable partial dentures is not optimal, with metal clasps visible on the buccal side of the teeth [2]. The novel permeation of high-performance polymer polyetheretherketone (PEEK) as a metal-free alternative material offers a new perspective for prosthetic applications [1].

PEEK is a semicrystalline linear polycyclic aromatic polymer, developed in 1978. After two, PEEK became an important high-performance thermoplastic material in orthopedic surgery [3]. Later, PEEK was introduced into the field of prosthetic dentistry. Because PEEK is increasingly used in the field of dentistry, its stability in the oral cavity must be taken into consideration. Previous studies reported that PEEK remains stable in nearly all organic and inorganic chemicals, and many studies demonstrated that PEEK can persist in many complex aging environments. However, an accurate understanding of the long-term performance of PEEK and filled PEEK in the oral cavity environment is essential [4].

PEEK is a white, radiolucent, rigid material with great thermal stability up to 335.8 °C. It is non allergic and has low plaque affinity. Flexural modulus of PEEK is 140–170 MPa, in addition to a lower density of 1.32 g/cm^3^ and thermal conductivity 0.29 W/mK. Young’s modulus of PEEK is relatively high, 3–4 GPa, close to human bone, enamel and dentin. PEEK is resistant to hydrolysis, non-toxic and has one of the best biocompatibilities compared to other materials. The special chemical structure of PEEK exhibits stable chemical and physical properties [3]. PEEK has the lowest solubility and water absorption values compared to PMMA and composites, a good chemical resistance against almost all solvents with almost zero water uptakes compared to alloys and ceramics, and exceptional tribological properties [1,2,3,4,5,6,7,8,9,10,11,12,13,14,15,16,17,18,19,20,21,22,23,24,25,26,27,28,29,30,31,32,33,34,35,36,37,38].

Despite high fracture resistance, PEEK is relatively weak mechanically in homogenic form. Clasps made of PEEK have lower strength than those made from metal alloys. Modified materials with fillers were developed in order to improve PEEK’s mechanical properties. PEEK containing 20% ceramic fillers known as BioHPP is non allergic and has high biocompatibility. Possibility of corrections, excellent stability, and great optimal polishability of BioHPP help to produce high-quality prosthetic restorations. It is a good alternative to Cr-Co frameworks for patients with high aesthetic requirements or allergic reactions [3,5]. Thus, using PEEK, metal-free and light weight frameworks can be produced as an alternative to conventional prosthetic restorations [1].

Unmodified PEEK is hydrophobic, having a contact angle of 80–90°. The addition of fillers can acquire hydrophilic characters [6].

Color stability is a significant clinical characteristic for dental restorations. Any color changes from the basic standard indicate aging or material damage. The color changes of dental materials during clinical use are caused by stain accumulation, dissolution of material ingredients, water absorption, pigment degradation, and surface roughness [7,8,9,10,11].

Likewise different compositions of the materials, microstructure, surface topography, and polishing procedures have been reported to increase or decrease vulnerability to discolorations [12,13]. Rough surfaces are more likely to stain than smooth surfaces, limiting also patient comfort. The roughness threshold (RT) for dental restorations is 0.2 μm for average roughness (Ra) [7].

The study was undertaken to evaluate the surface characteristics, microhardness, and color stability of PEEK materials related to water saturation and in vitro aging. The primary first hypothesis set for the study was that there is no direct correlation between the amount of water sorption and the reduction in hardness. The secondary null hypothesis was that surface characteristics are affected by water immersion and thermocycling. The third null hypothesis was that there would be marked color changes after water immersion and in vitro artificial aging.

## 2. Materials and Methods

### 2.1. Specimen Preparation

Custom specimens of unmodified and modified PEEK CAD/CAM materials were investigated. The manufacturers, material types, and compositions are listed in Table 1. Forty-eight plates were sectioned in rectangular slices (15 × 10 × 1 mm^3^) (n = 16) using an equipment with microns precision in order to provide standardization for the samples. The specimens were grinded and polished using #600–2000 silicon carbide paper. The final thickness of each slice was checked with a digital caliper. The specimens were finally polished with a low-speed handpiece and diamond polishing paste Renfert polish (Renfert, Hilzingen, Germany), were ultrasonically cleaned for 10 min, and were air-dried. They were weighed on an analytical balance, accurate to 0.00001 g. Each sample was weighted one time.

### 2.2. Water Saturation

The specimens were immersed in distilled water at 37 °C. They were removed from water after seven days, wiped gently until free from visible moisture, and weighed 60 s after removal until constant mass was reached, called water saturation. The specimens were re-immersed in water and the measurement procedure repeated every 7 days, for a period of 28 days in all. They were weighed on an analytical balance AS 220.R2 PLUS (Synergy Lab line-RADWAG, Radom, Poland) with five decimals (0.00001) after each period.

### 2.3. Thermal Cycling

The specimens were then subjected to aging by thermal cycling. A total of 10,000 cycles in the water bath (5–55 °C, 30 s dwell time and 10 s transfer time) was applied to simulate one year of oral use. Plates were weighted again on an analytical balance [39,40]. This protocol was selected based on the standard ISO number 11,405 and simulates the temperature changes that take place in the oral environment.

### 2.4. Surface Roughness Measurements

Surface roughness was analyzed with a contact 2 µm stylus profilometer Surftest SJ-201 (Mitutoyo, Kawasaki, Japan) before water immersion, after each immersion period and after aging, for each group. Arithmetic average roughness (Ra) and maximum absolute vertical roughness (Rz) measurements were performed in 5 different directions and all data were recorded. The mean value of the five measurements was calculated for each surface. The sampling length was 0.3 mm, and a force of 0.7 mN was applied. Measurements were made on dry specimens before immersion in water and then subsequently once a week for 4 weeks and after thermocycling.

### 2.5. Nanosurface Topographic Characterization by Atomic Force Microscopy (AFM) 

Samples from each group were examined with an atomic force microscope Nanosurf Easy Scan 2 Advanced Research (NanosurfAG, Liestal, Switzerland), in non-contact mode and values for average nanoroughness Sa (nm), maximum amplitude of heights Sy (nm) were registered. AFM generated corresponding three-dimensional profiles of the sample surfaces (1.12 µm × 1.12 µm). AFM is a cantilever-based technique that utilizes a sharp tip to interrogate surfaces at resolutions below the optical diffraction limit. It is also a powerful tool for nano-mechanical probing and measurements.

Measurements were made on dry specimens before immersion in water and then subsequently once a week for 4 weeks, and after thermocycling.

### 2.6. Hardness Measurements

Vickers hardness measurements were performed with a micro-hardness tester DM 8/DM 2 (Yang Yi Technology Co., Lt, Tainan City 70960, Taiwan, China) using a load of 300 g applied for 10 s with a diamond pyramidal indenter. After removing the indenter, the indentation dimensions were microscopically recorded at a 40× magnification. Hardness measurements were made on selected points. Five indentations were made on each specimen. Measurements were made on dry specimens before immersion in water, after each week, during four weeks, and after thermocycling. Hardness values were calculated with the following Formula (5):HV = 1.8544 F/d^2^(1)
where HV is the Vickers hardness number, F is the load, and d is the indentation diagonal length.

### 2.7. Optical and Color Changes Measurements 

Translucency (TP), opalescence parameters (OP), and contrast ratio(CR)were determined for all specimens on dry specimens before immersion in water and then subsequently once a week for 4 weeks, and after thermocycling. Optical properties were calculated under a D65 illuminant, using a spectrophotometer Vita Easyshade IV (Vita Zahnfabrick, Bad Säckingen, Germany). The spectrophotometer was calibrated before each measurement.

A black (b) and white (w) background were used to assess the measurements, using a grey card WhiBal G7 (White Balance Pocket Card). L* is a measure of the lightness-darkness of material (perfect black has an L* = 0, and perfect white has an L* = 100). a* coordinate represents the redness (positive value) or the greenness (negative value), while the b* coordinate is a measure of the yellowness (positive value) or the blueness (negative value) [14,15,16,17].

TP values were calculated using the Equation (2).
TP = [(L_b_ − L_w_)^2^ + (a_b_ − a_w_)^2^ + (b_b_ − b_w_)^2^]^1/2^(2)

OP values were calculated using the Equation (3).
OP = [(a_b_ − a_w_)^2^ + (b_b_ − b_w_)^2^]^1/2^(3)

CR was achieved by Equation (4).
CR = Y_b_/Y_w_ Y = [(L* + 16)/116]^3^ × 100(4)
where w and b are color coordinates of the specimens on the white and black backgrounds. In this calculation, CR = 0 is considered transparent, and CR = 1 is regarded as totally opaque [18].

The color changes (ΔE*) were calculated based on the CIE L*a*b* color system. L* represents lightness (+bright, and −dark); a* represents the color scale from red (+) to green (−) and b* the color scale from yellow (+) to blue (−).

The total color change value (ΔΕ*) was calculated according to Equation (5), which represents the color difference before and after immersion.
ΔΕ* = [(ΔL*)^2^ + (Δa*)^2^ +(Δb*)^2^]^1/2^(5)

Measurements were made for each group.

The national bureau of standards (NBS) system was used to quantify the levels of color change (Table 2). To relate the color change to a clinical standard, the ΔE* values were converted into NBS units: NBS = ΔE* × 0.92 [19,20,21,22,23].

### 2.8. Statistical Analysis

Statistical analyses were performed by means of the IBM SPSS Statistics software (IBM, New York, NY, USA). Differences among the variables were made. Average values and standard deviations (SD) were calculated. Paired t test was used to evaluate the comparisons between the means. The level of significance was set to 0.05. Spearman correlation was used to assess relationships between variables. It measures the strength of association between variables and the direction of the relationship. The significance was related to: 00–0.19 “very weak”, 0.20–0.39 “weak”, 0.40–0.59 “moderate”, 0.60–0.79 “strong”, and 0.80–1.0 “very strong”.

## 3. Results

### 3.1. Water Saturation

After the first week of immersion in distilled water at 37 °C, the weight of all specimens increased by0.21–0.27%, for all materials. Otherwise, the weight changes in subsequent weeks were lower than 0.05% (Figure 1). After thermocycling the weight of the samples still decreased.

Statistical analyses registered significant differences (*p* < 0.05) between the weight of the samples before and after the first week of immersion. The weight changes during the investigations are significantly different between the three investigated materials (*p* < 0.05). The reinforced PEEK absorbs water at least during immersion, but during thermocycling loses most of the weight (0.02%). 

### 3.2. Roughness Measurements

Registered mean roughness values were 0.092 ± 0.008 µm for B, 0.083 ± 0.008 µm for F, and 0.080 ± 0.008 µm for J before water immersion. They increased significantly (*p* < 0.05) after the first week of water immersion, to 0.113 ± 0.009 µm for B, 0.106 ± 0.009 µm for F, and 0.101 ± 0.009 µm for J, and decreased significantly after the following second week, becoming 0.097 ± 0.009 µm for B, 0.093 ± 0.009 µm for F, and 0.095 ± 0.009 µm for J. Starting with the third week, the roughness variations become insignificant.

Thermocycling induces a significant increase of microroughness, becoming 0.103 ± 0.010 µm for B, 0.109 ± 0.009 µm for F and 0.108 ± 0.010 µm for J (Figure 2).

Related to the maximum absolute vertical roughness values, they vary in the same way.

The roughness behavior of the three investigated materials is statistically not different.

### 3.3. Nanosurface Topographic Characterization by Atomic Force Microscopy (AFM) 

AFM characterization offers both qualitative and quantitative information including high resolution nanoscale images for the morphology, surface texture, particle size, and roughness. The 3D images of the samples before water immersion show irregular surfaces, with depressions of different sizes, rounded edges. The depth of the depressions decreases as follows: B > F > J. Due to the very small grain size of B, constant homogeneity can be produced. During water immersion, the 3D surfaces change very little in appearance, so that after the last week of immersion in water the ratio between materials changes as: F > J > B. After thermocycling, it returns to the initial report, but the depressions decrease (Figure 3). 3D images correlate with the average particle size (Figure 4).

Related to the mean particle size, before immersion, the values are B > F > J. For B, the values decrease during water immersion and after tc. For F and J, they increase during water immersion and decrease after tc.

Nanoroughness values (Figure 5) increase after immersion during the first two weeks, then increase further only for F, and decreasefor B and J. In relation to tc, they decrease for B and F and increase for J.

There is a moderate correlation (R = 0.49) between Ra and Sa, related to tc. The most affected by tc are F and J. The reinforced PEEK material is the least modified by aging, related to the nanosurface topography and nanoroughness.

### 3.4. Hardness Measurements

The hardness values before water immersion, during the 4 weeks, and after tc are shown in Table 3 and Figure 6. They decrease continuously. The differences are significant during the first and second weeks (*p* = 0.006, and *p* = 0.019), and then, the differences become insignificant. The hardness decrease is significant during the whole period of immersion and during immersion associated with thermocycling. The hardness decreases during water immersion by 10.94% for B, 10.89% for F, and 7.23% for J. After thermocycling, the hardness decreases still by 2.31% for B, 0.12% for F and 1.81% for J. Another aspect is that for B and F, the biggestdecrease is in the first week and for J in the second. B and F behave similarly during immersion (*p* = 0.129). Otherwise, during tc and related to the whole hardness decrease behavior, there are significant differences between all materials (*p* < 0.05). In relation to microhardness, the most affected by water immersion and thermocycling is B, followed by F, and finally J.

### 3.5. Optical and Color Changes Measurements 

After water immersion and thermocycling TP and OP parameters decrease and CR increases. The modifications are higher for OP. In relation to the materials, before aging, the TP and OP increase in the order B < F < J and after aging in the order F < J < B. In relation to the materials, there are not significant differences in terms of TP parameters, but related to OP, the behavior of J is significantly different (*p* = 0.029 for B–J, and *p* = 0.039 for F–J), the opalescence values decrease more slowly for J, during water immersion (Figure 7).

In relation to the levels of color change, according to the NBS (Table 4), after 4 weeks of water immersion, they are between extremely slight and slight for all materials. Thermocycling induces extremely slight changes for B and J and slight for F. B is more changed by immersion and F by tc. 

## 4. Discussion

A significant characteristic of dental polymers is to absorb water that diffuses through the resin matrix and for composites probably also through the filler/matrix interfaces [4,24].

A plasticization of the polymer occurs, which produces softening and deterioration of the surface properties and consequently of the optical characteristics of the material. Different studies demonstrated that even the crystalline regions of the semicrystalline polymers act as tie points to inhibit the dilation of the polymer and consequently its subsequent plasticization and water uptake induce a considerable decrease in the properties of the semicrystalline PEEK [25].

Studies show that most of the resin materials absorbed water and started to reach saturation after 7 days of immersion. Water plasticization can be maximized after reaching saturation, resulting in polymer chain separation and hardness reduction [26]. There seems to be no direct correlation between the amount of water sorption and the reduction in hardness [2].

The current study revealed that all materials reached saturation after the first week of immersion and during in vitro aging they lose weight. Water absorption was associated with a decrease in microhardness, but the decrease continues even after the first week of immersion. This fact supports a moderate direct correlation (R = 0.44) between water sorption and microhardness decrease. Therefore, the first null hypothesis is rejected. In relation to microhardness, the PEEK most affected by water immersion and thermocycling is the reinforced PEEK. Microhardness values are lower than inother dental materials such as Co-Cr.

During aging, the clinical performance of the materials could be altered by changes in their morphology and optical properties. The degradation of PEEK and reinforced PEEK in the oral cavity has not been sufficient investigated. There is little information in the current literature on the impact of thermal cycling on a material’s surface properties and color changes. 

Even the nanohardness is increased in reinforced PEEK, and the material becomes a complex system, where the properties are not a sum of the properties of the components but are influenced by the characteristics of the interfacial area between the fillers and the matrix. Previous studies show that the nanomechanical properties of PEEK were not affected by immersion in artificial saliva at physiological temperatures over 30 days by the absorption of solution and that a simulative oral environment caused no significant changes inPEEK materials. However, the stability of nanomechanical properties of reinforced PEEK was more affected than unfilled PEEK after the aging treatment [4].

The PEEK with 20% ceramic filler exhibited the highest hardness reduction in artificial saliva and this might be due to the interaction between filler and matrix. PEEK has very stable chemical and physical properties due to the presence of an aryl ring containing ketone and other groups. Consequently, it has resistance to surface modification; it is stable at high temperatures and has shown low solubility and water sorption compared to other polymer-based CAD/CAM materials [27,28].

The chemical composition, the hydrophilic characteristics of these monomers, and polymer network configurations are also important factors that determine composite performance, when exposed to water artificial aging [29].

Thermocycling can be applied as a method of artificial aging, which mainly consists of water immersion and temperature change under standardized laboratory conditions [30]. It can be assumed that thermal cycling leads to microcracks, which increase the surface roughness and enhance water uptake and dissolution [31].

The current study demonstrated that water immersion and thermocycling induced a significant increase of microroughness, with no significant differences between materials. However, the roughness is kept within clinically accepted limits. A moderate correlation (R = 0.49) was found between Ra and Sa, related to tc. The reinforced PEEK material is the least modified by aging, related to the nanosurface topography and nanoroughness. The secondary null hypothesis is accepted; surface characteristics of the materials are affected by water immersion and thermocycling.

Polymer networks tend to absorb water, which can lead not only to mechanical weakening due to microstructural changes but also to higher optical transformations, with an aesthetically compromised outcome [13,32,33,34,35,36].

However, this study revealed only extremely slight and slight color changes of the materials, during water immersion and artificial aging, even if increased values of the translucency and opalescence parameters were measured. Therefore, the third null hypothesis of the study could be rejected. 

The PEEK material can be considered promising as a future alternative to metals, due to its high-quality mechanical properties [37,38,39,40]. The modified PEEK materials have shown better properties than the unmodified form of PEEK. The modified PEEK contains ceramic fillers that improve the properties of the materials. This study revealed a better aging behavior of the reinforced PEEK in terms of surface characteristics. Due to its elastic modulus, strength, rigidity, and lightweight, the applications of PEEK are for different clinical situations in dental practice. More research is needed on PEEK polymer as an alternative material for existing metals which have been used for a long time [6].

It is important to bear in mind that in this study only a limited number of materials were investigated and only few factors were taken into account. Thus, further long-term studies should investigate the aging behavior of dental PEEK materials related to the complex oral environment.

## 5. Conclusions

Within the limits of this study, the following conclusions are drawn:The studied reinforced and unfilled PEEK reached water saturation after the first week of immersion, without significant differences between them. Water absorption was associated with a decrease in microhardness. The most affected from this point of view was the reinforced PEEK material.Thermocycling induced a significant increase of microroughness, without significant differences between the studied materials.Related to the nanosurface topography and roughness, the reinforced PEEK material was the least modified by aging.The color changes after 4 weeks of water immersion and one year of simulated in vitro aging ranged from extremely slight to slight, for all materials.

## Figures and Tables

**Figure 1 polymers-14-02144-f001:**
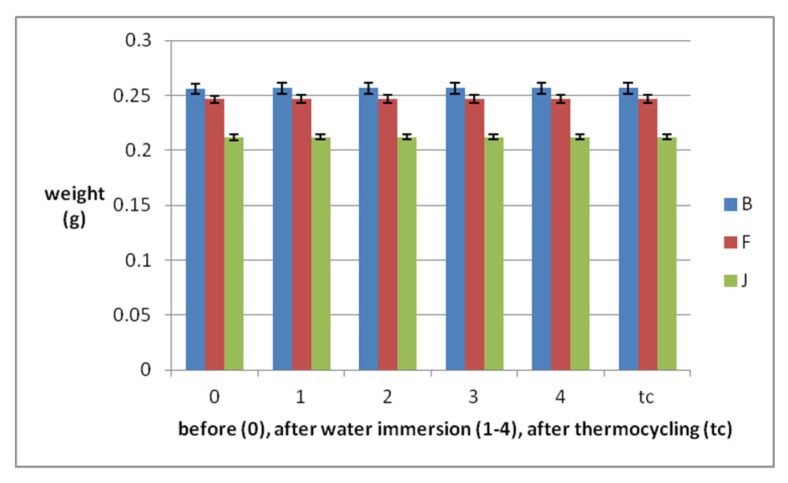
Weight variations of the samples during water immersion periods and thermocycling: average weight values with SD.

**Figure 2 polymers-14-02144-f002:**
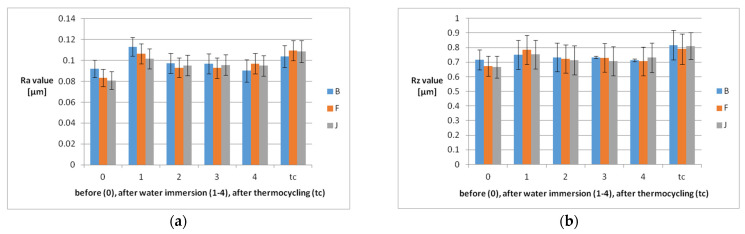
Mean roughness values of the samples with SD before (0), after each week of water immersion (1–4), and after thermocycling: (**a**) Ra values and (**b**) Rz values.

**Figure 3 polymers-14-02144-f003:**
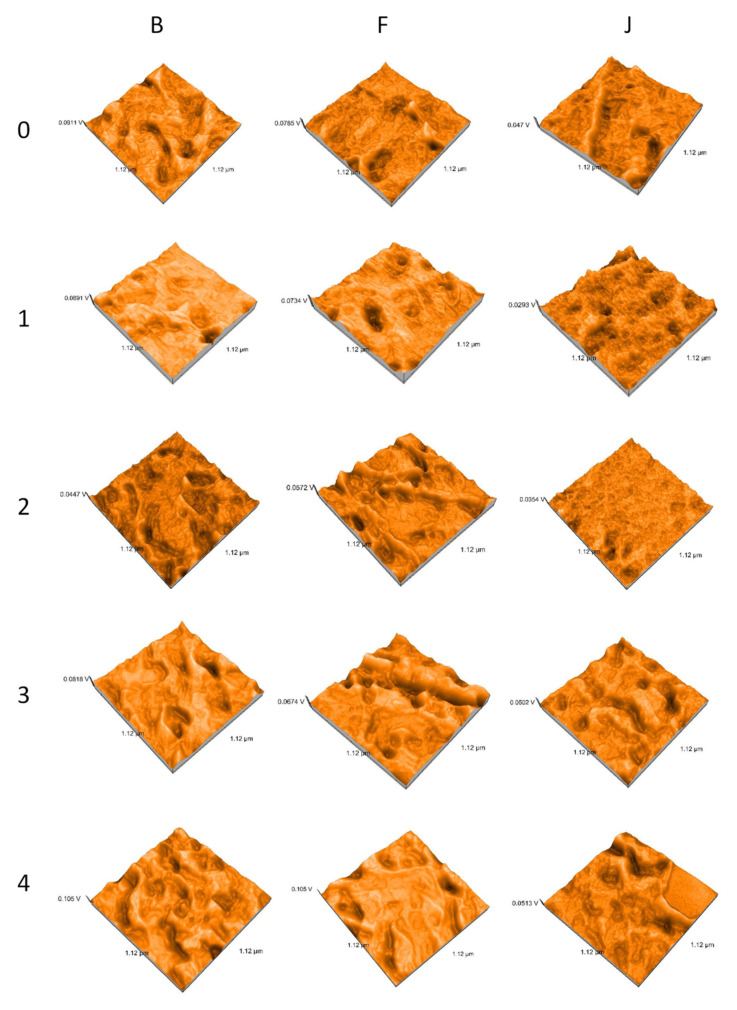
3D reconstructions of AFM images.

**Figure 4 polymers-14-02144-f004:**
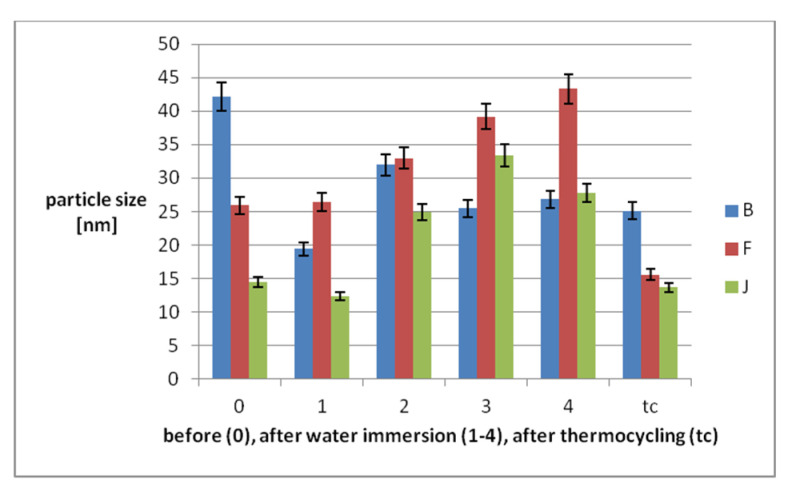
Mean particle size on the materials surfaces.

**Figure 5 polymers-14-02144-f005:**
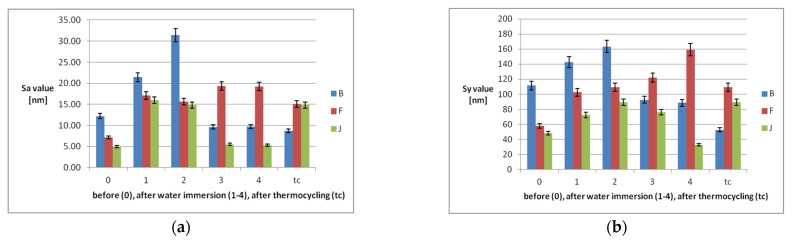
Nanoroughness values of the samples before each week of water immersion(0), after (1–4), and after thermocycling: (**a**) Sa values and (**b**) Sy values.

**Figure 6 polymers-14-02144-f006:**
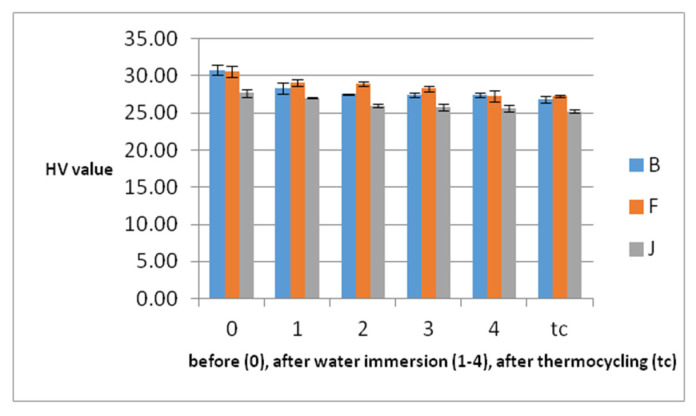
Mean hardness values of the samples with SD before each week of water immersion (0), after (1–4), and after thermocycling.

**Figure 7 polymers-14-02144-f007:**
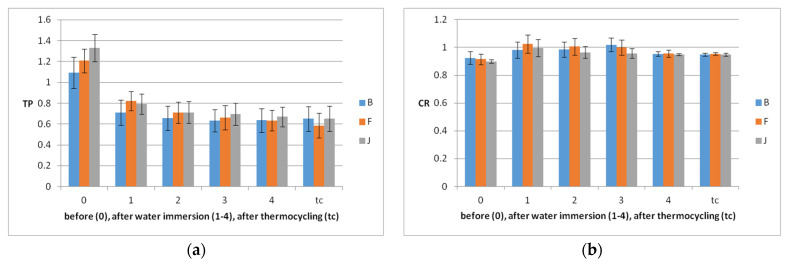
Mean optical parameters values of the samples with SD before each week of water immersion (0), after (1–4), and after thermocycling: (**a**) TP, (**b**) CR, and (**c**) OP.

**Table 1 polymers-14-02144-t001:** PEEK materials taken into the study.

Material	Composition	Manufacturer
BioHPP dentine shade 2 (B)	Ceramic reinforced (grain size of 0.3 to 0.5 μm) partially crystalline polyetheretherketone (PEEK)	Bredent group GmbH & Co. KG, Senden, Germany
Finoframe PEEK A2 (F)	Polyetheretherketone (PEEK) 100%	Fino GmbH, Bad Bocklet, Germany
Juvora medical PEEK nature (J)	Polyetheretherketone (PEEK) 100%	Juvora Ltd. Global Technology Center, Lancashire, UK

**Table 2 polymers-14-02144-t002:** Levels of color change, according to NBS.

NBS Units	Color Changes
0.0–0.5	extremely slight change
0.5–1.5	slight change
1.5–3.0	perceivable
3.0–6.0	marked change
6.0–12.0	extremely marked change
12.0 or more	change to another color

**Table 3 polymers-14-02144-t003:** Mean hardness values of the samples before water immersion, during the 4 weeks, and after tc with SD.

Material	Before	After 1 Week	After 2 Weeks	After 3 Weeks	After 4 Weeks	After tc
B	30.77 ± 0.67	28.37 ± 0.77	27.50 ± 0.08	27.4 ± 0.30	27.40 ± 0.35	26.76 ± 0.44
F	30.60 ± 0.77	29.03 ± 0.44	28.93 ± 0.33	28.26 ± 0.33	27.27 ± 0.68	27.23 ± 0.12
J	27.67 ± 0.52	27.03 ± 0.12	25.97 ± 0.24	25.76 ± 0.38	25.67 ± 0.44	25.20 ± 0.21

**Table 4 polymers-14-02144-t004:** Color changes of the materials, on black (b) and white (w) backgrounds, related to each stage of immersion and during thermocycling.

Color Change	B(b)	B(w)	F(b)	F(w)	J(b)	J(w)
0-1	0.506	0.242	0.217	1.462	0.646	1.059
0-2	1.293	0.804	1.105	1.319	0.685	0.341
0-3	1.002	1.207	0.366	1.344	0.724	0.341
0-4	1.383	1.075	0.613	0.852	0.737	0.312
4-tc	0.437	0.420	0.544	0.594	0.053	0.101

## Data Availability

Not applicable.

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
