# Peer review of "Surface Characteristics and Color Stability of Dental PEEK Related to Water Saturation and Thermal Cycling"

_polymers, 2022, doi:10.3390/polym14112144_

Round 1

Reviewer 1 Report

  1. PMMA is a commonly used material in dentistry. But I haven't seen any explanation in this article that mentions PEKK's better advantages over PMMA, please explain.

  2. According to the first point of the conclusion, can it be determined that the reinforced material is not a better choice?
  3.  The screen resolution of AFM is not good, please improve it.
  4. Please provide reference materials for the selection of the number of times (10,000 cycles) and the selection of temperature (5 degrees to 50 degrees) in the cooling and heating cycle test, and explain why such experimental conditions are selected.
  5. All materials will change color after soaking in water for four weeks. Whether the clinical application value can be considered that the material has low applicability

Author Response

We thank You for the review!

  • PMMA is a commonly used material in dentistry. But I haven't seen any explanation in this article that mentions PEKK's better advantages over PMMA, please explain.
    Response: We agree to this point. Despite the fact, that both materials, PMMA and PEEK are thermoplastics, their properties are very different. PEEK material is a high-performance polymer semi-crystalline thermoplastic with excellent mechanical strength compared to PPMA, which is not high performance polymer.
  • According to the first point of the conclusion, can it be determined that the reinforced material is not a better choice?

Response: Based on the results on this study, there was significant changes regarding this material, but this is a good choice.

  • The screen resolution of AFM is not good, please improve it.

Response: We agree to this point and we changed the figures.

  • Please provide reference materials for the selection of the number of times (10,000 cycles) and the selection of temperature (5 degrees to 50 degrees) in the cooling and heating cycle test, and explain why such experimental conditions are selected.

Response: We agree to this point and added new references in text.

This protocol was selected because it simulates the oral environment.

 Ghavami-Lahiji, Mehrsima et al. “The effect of thermocycling on the degree of conversion and mechanical properties of a microhybrid dental resin composite.” Restorative dentistry & endodontics vol. 43,2 e26. 26 Apr. 2018, doi:10.5395/rde.2018.43.e26

Kiomarsi, Nazanin et al. “Effect of thermocycling and surface treatment on repair bond strength of composite.” Journal of clinical and experimental dentistry vol. 9,8 e945-e951. 1 Aug. 2017, doi:10.4317/jced.53721

Atalay S, Çakmak G, Fonseca M, Schimmel M, Yilmaz B. Effect of thermocycling on the surface properties of CAD-CAM denture base materials after different surface treatments. J Mech Behav Biomed Mater. 2021 Sep;121:104646. doi: 10.1016/j.jmbbm.2021.104646. Epub 2021 Jun 16. PMID: 34166873.

  • All materials will change color after soaking in water for four weeks. Whether the clinical application value can be considered that the material has low applicability.

Response: This is correct, that all materials can change color in water after this amount of time, but we wanted to quantify the degree.  

Reviewer 2 Report

This manuscript describes an experimental study comparing the surface characteristics, mechanical properties, and color stability of original/modified PEEK materials. These properties are clinically important as they may affect the material's longevity and esthetics. However, in the reviewer's opinion, the manuscript is not suitable for publication in the current form due to several conceptual and technological issues.

Major comments:

  1. One major issue is that the authors did not follow any international, state, or other industrial standards for the evaluation of dental material (including specimen preparation and testing), making data interpretation difficult. 
  2. Due to the same reason above, it is unclear whether the difference reported between different time points or different materials are clinically significant (although the values may be statistically different). 
  3. To address the issues and to make the experimental assessment meaningful, I suggest that the authors carefully revisit those industrial standards available, and reevaluate the methods selected. In my opinion, the standards used should be clearly stated in the manuscript and strictly followed.

Minor comments:

  1. If Finoframe PEEK A2 and Juvora medical PEEK nature are both 100% (unmodified) PEEK, I wonder why they behave so differently. Also, if unmodified PEEK materials themselves are so different, do you still think the “unique features” of the reinforced PEEK (e.g., more affected by water absorption and aging) are actually caused by the modification (addition of fillers)?
  2. General comments for the figures: I find the figures hard to understand - the coding for the groups (0,1,2,3,4) is confusing. It may be more clear to just use “baseline”, “week-1”, “week-2” etc.; Also, It may be helpful to annotate the result of statistical analysis in the graphs; More detailed figure caption/legend could also help; Standard deviations are missing in several figures.
  3. Figure 3 is very unclear. The particles are barely visible from the image and the text is not legible. There is little consistency between Fig. 3 and Fig. 4, although the authors claimed that data in Fig.4 were based on the AFM analysis in Fig.3.
  4. The authors should revise the manuscript responsibly before submission. Contents from the template of the journal are still in the manuscript, e.g., at the beginning of the Conclusions and Author Contributions.

Author Response

Thank You very much for Your review!

This manuscript describes an experimental study comparing the surface characteristics, mechanical properties, and color stability of original/modified PEEK materials. These properties are clinically important as they may affect the material's longevity and esthetics. However, in the reviewer's opinion, the manuscript is not suitable for publication in the current form due to several conceptual and technological issues.

Major comments:

One major issue is that the authors did not follow any international, state, or other industrial standards for the evaluation of dental material (including specimen preparation and testing), making data interpretation difficult.

Response: We agree to this point. The thermal aging process is standardized according to ISO 11405. The specimen preparation was done respecting the protocols found in research literature. The specimens were prepared respecting the manufacturer’s indications.

Due to the same reason above, it is unclear whether the difference reported between different time points or different materials are clinically significant (although the values may be statistically different).

To address the issues and to make the experimental assessment meaningful, I suggest that the authors carefully revisit those industrial standards available, and reevaluate the methods selected. In my opinion, the standards used should be clearly stated in the manuscript and strictly followed.

Minor comments:

If Finoframe PEEK A2 and Juvora medical PEEK nature are both 100% (unmodified) PEEK, I wonder why they behave so differently. Also, if unmodified PEEK materials themselves are so different, do you still think the “unique features” of the reinforced PEEK (e.g., more affected by water absorption and aging) are actually caused by the modification (addition of fillers)?

Response: We think that could be an explanation for the different behaviour of the materials included in this manuscript.

General comments for the figures: I find the figures hard to understand - the coding for the groups (0,1,2,3,4) is confusing. It may be more clear to just use “baseline”, “week-1”, “week-2” etc.; Also, It may be helpful to annotate the result of statistical analysis in the graphs; More detailed figure caption/legend could also help; Standard deviations are missing in several figures.

Response: We agree to this point and added standard deviations to the figures.

Figure 3 is very unclear. The particles are barely visible from the image and the text is not legible. There is little consistency between Fig. 3 and Fig. 4, although the authors claimed that data in Fig.4 were based on the AFM analysis in Fig.3.

Response: We  agree to this point and we changed the figures in order to be clearly read.

The authors should revise the manuscript responsibly before submission. Contents from the template of the journal are still in the manuscript, e.g., at the beginning of the Conclusions and Author Contributions.

Response: We agree to this point and corrected.

Reviewer 3 Report

This is an interesting study looking at various properties of PEEK for use in dental applications and how the properties change when the material uptakes water and undergoes thermal cycling.  This type of study is important to better understand the suitability of PEEK for dental applications.

Abstract

Line 34 – please change ‘extra slight’ to ‘extremely slight’ as per the table

Introduction

Paragraphs 4 and 5 – regarding the use of terms such as ‘best’ or ‘lowest’ when talking about PEEK.  When these terms are used, a comparison needs to be made ideally with the industry standard.

When talking about PEEK, its biocompatibility and mechanical properties (paragraph 4), consider articles where PEEK has been used as a biomaterial in other applications eg, Kurtz SM, Devine JN. PEEK biomaterials in trauma, orthopedic, and spinal implants. Biomaterials. 2007 Nov;28(32):4845-69. 

Line 85 – consider changing ‘sorption’ to ‘absorption’ as it is the moisture uptake by the polymer being measured

Line 90 – please clarify the roughness.  Is 0.2 and Ra value?  Please state.

Materials and Methods

Table 1 – Please add details of the ceramic reinforcements used, what is the composition of the ceramic, are the grains randomly distributed through the PEEK?  Are there differences between the two PEEK materials used?

Section 2.1 describe the drying of the components

For weighing the components (section 2.2), please describe how many times each sample was weighed and the specific weighing protocol used.

Section 2.3 – is the thermal cycling protocol a standard protocol or has it been used previously?  Can it be referenced?

Section 2.5 – how many AFM measurements were taken?

Section 2.7 – Please define TP, OP and CR or ideally remove these acronyms as they make the article less accessible to the reader

Results

Figure 1 – Please avoid acronyms in the graphs, the B, F and J notation in both the graphs and text make reading the manuscript difficult and add to confusion, please consider writing in full in both the graphs and the text.  Please label the x axes fully, in figure 1(a); the notation used in figure 1(b) is more clear.  Figure 1 a and b show the same information, please remove one of these graphs so that the same data is not repeated in the manuscript.  Please include n numbers in all the legends.

Figure 2 – could the colours of the bars be kept consistent for the different materials throughout?  See comments for figure 1 about the use of acronyms and the labelling on the x-axis

Figure 3 – could these images be made larger?  The scale bars are difficult to read.  Again, please label the columns of the table properly, 0, 1, 2… are confusing.

Figure 4 and 5 – Please include error bars on these graphs and clarify the x axes

Table 1 and Figure 6 and 7 – please do not repeat the data, either remove the table or the graph and figure 7.

Section 3.5 – please write TC, OP and CR in full, this would help readers less familiar with the subject

Discussion

Please comment on whether water uptake influences the geometry of the materials

Please comment on whether any of the changes particularly in the mechanical properties were great enough to induce failure of the denture.  Were the colour and surface roughness changes acceptable?

Please comment on the moisture uptake of PEEK and the potential errors in the weighing, previous studies looking at the use of PEEK as an arthroplasty bearing material have reported extended durations of soaking being required for the material to stabilise eg. Brockett, CL., et al. "Wear of ceramic‐on‐carbon fiber‐reinforced poly‐ether ether ketone hip replacements." Journal of Biomedical Materials Research Part B: Applied Biomaterials 100.6 (2012): 1459-1465. and Cowie, RM., et al. "Wear and Friction of UHMWPE-on-PEEK OPTIMA™." Journal of the Mechanical Behavior of Biomedical Materials 89 (2019): 65-71.  How do the measurements taken in this study compare to previous investigations where rigorous cleaning, drying and weighing protocols have been used?

For the nanohardness measurements, please comment on how the values obtained in this study compare to conventional denture materials eg. CoCr.  Would a lower hardness result in scratching of he material?  Would this lead to material loss?  What influence would this have on the aesthetics and cleaning of the denture?  Scratching and third body wear of PEEK has been investigated for orthopaedic implant materials.  PEEK has been shown to scratch easily but when articulating against polyethylene has been shown to have a self-polishing effect (Cowie, R M., et al. "Third body wear of UHMWPE-on-PEEK-OPTIMA™." Materials 13.6 (2020): 1264.)

Line 447 – please change ‘extra slight’ to ‘extremely slight’ as per the table

Line 482 – please elaborate on the ‘modified’ PEEK.  What materials were used in the modification?

There are differences between the two 100% PEEK materials.  Do the findings from this study translate to all PEEK materials?  Are the differences due to the batch or manufacturer?

How do the findings from this study influence how PEEK is used in dentistry?  Eg. Should colour matching and manufacture be carried out on pre-soaked material?

What is the intended processing route for the materials when used in dentistry?  Will the manufacturing process influence the crystallinity of the material particularly on the surface as the rate of cooling of PEEK has been shown to influence its crystallinity and therefore colour.

Conclusions

Please remove the first sentence of the conclusion (line 493-494)

Line 505 – please change ‘extra slight’ to ‘extremely slight’ as per the table

Additional typos

Line 28 – Change ‘Studied’ to ‘The studied’?

Line 30 – change ‘induce’ to ‘induces’

Line 42 – Change ‘strong’ to ‘strongly’

Line 121 – change ‘weight’ to ‘weighed’

Line 452 – change ‘nanomechanic’ to ‘nanomechanical’

Line 489 – change ‘was’ to ‘were’

Line 496 – Change ‘Studied’ to ‘The studied’?

Author Response

This is an interesting study looking at various properties of PEEK for use in dental applications and how the properties change when the material uptakes water and undergoes thermal cycling.  This type of study is important to better understand the suitability of PEEK for dental applications.

Abstract

Line 34 – please change ‘extra slight’ to ‘extremely slight’ as per the table

Response: We agree to this point and corrected.

Introduction

Paragraphs 4 and 5 – regarding the use of terms such as ‘best’ or ‘lowest’ when talking about PEEK.  When these terms are used, a comparison needs to be made ideally with the industry standard.

Response: We agree to this point and added other information.

When talking about PEEK, its biocompatibility and mechanical properties (paragraph 4), consider articles where PEEK has been used as a biomaterial in other applications eg, Kurtz SM, Devine JN. PEEK biomaterials in trauma, orthopedic, and spinal implants. Biomaterials. 2007 Nov;28(32):4845-69.

Response: We agree to this point and added the reference in the manuscript.

Line 85 – consider changing ‘sorption’ to ‘absorption’ as it is the moisture uptake by the polymer being measured

Response: We agree to this point and changed the term.

Line 90 – please clarify the roughness.  Is 0.2 and Ra value?  Please state.

Response: We agree to this point and added new information.

‘’Likewise different compositions of the materials, microstructure, surface topogra-phy and polishing procedures have been reported to increase or decrease vulnerability to discolorations [12, 13]. Rough surfaces are more likely to staining than smooth surfaces, limiting also patient comfort. The roughness threshold (RT) for dental restorations is 0.2 μm for average roughness (Ra)[7].’’

Materials and Methods

Table 1 – Please add details of the ceramic reinforcements used, what is the composition of the ceramic, are the grains randomly distributed through the PEEK?  Are there differences between the two PEEK materials used?

Response: We agree to this point, but the manufacture doesn’t provide more information about the ‘’special ceramic filler’’.

Section 2.1 describe the drying of the components

Response: We agree to this point and the samples were air-dried.

For weighing the components (section 2.2), please describe how many times each sample was weighed and the specific weighing protocol used.

Response: We agree to this point. Each samples was weighed one time.

Section 2.3 – is the thermal cycling protocol a standard protocol or has it been used previously?  Can it be referenced?

Response: We agree to this point and the protocol was used previously. We added references in the manuscript. According to ISO 11405, the use of 500 thermal cycles between 5°C and 55°C is considered to be suitable to simulate short-term aging of dental materials.

Ghavami-Lahiji, Mehrsima et al. “The effect of thermocycling on the degree of conversion and mechanical properties of a microhybrid dental resin composite.” Restorative dentistry & endodontics vol. 43,2 e26. 26 Apr. 2018, doi:10.5395/rde.2018.43.e26

Kiomarsi, Nazanin et al. “Effect of thermocycling and surface treatment on repair bond strength of composite.” Journal of clinical and experimental dentistry vol. 9,8 e945-e951. 1 Aug. 2017, doi:10.4317/jced.53721

Atalay S, Çakmak G, Fonseca M, Schimmel M, Yilmaz B. Effect of thermocycling on the surface properties of CAD-CAM denture base materials after different surface treatments. J Mech Behav Biomed Mater. 2021 Sep;121:104646. doi: 10.1016/j.jmbbm.2021.104646. Epub 2021 Jun 16. PMID: 34166873.

Section 2.5 – how many AFM measurements were taken?

Response: Each samples was measured multiple times in order to research the most significant area for this research.

Section 2.7 – Please define TP, OP and CR or ideally remove these acronyms as they make the article less accessible to the reader

Response: We agree to this point and added the acronyms for the parameters.

Results

Figure 1 – Please avoid acronyms in the graphs, the B, F and J notation in both the graphs and text make reading the manuscript difficult and add to confusion, please consider writing in full in both the graphs and the text.  Please label the x axes fully, in figure 1(a); the notation used in figure 1(b) is more clear.  Figure 1 a and b show the same information, please remove one of these graphs so that the same data is not repeated in the manuscript.  Please include n numbers in all the legends.

Response: We agree to this point and removed the figure b. Part of the acronyms were kept in order to be easily read.

Figure 2 – could the colours of the bars be kept consistent for the different materials throughout? 

Response: We agree to this point but we selected the different colours in order to highlight the materials in order to be easy to understand for the readers.

See comments for figure 1 about the use of acronyms and the labelling on the x-axis

Response:

Figure 3 – could these images be made larger?  The scale bars are difficult to read.  Again, please label the columns of the table properly, 0, 1, 2… are confusing.

Response: We agree to this point and changed the figure 3.

Figure 4 and 5 – Please include error bars on these graphs and clarify the x axes

Response: We agree to this point and added errors bars.

Table 1 and Figure 6 and 7 – please do not repeat the data, either remove the table or the graph and figure 7.

Response: We agree to this point and removed the Figure 7.

Section 3.5 – please write TC, OP and CR in full, this would help readers less familiar with the subject

Response: We agree to this point and we chose the acronyms in order to be easy to read the manuscript.

Discussion

Please comment on whether water uptake influences the geometry of the materials

Response: We found your point very interesting but the water uptake did not change the geometry of the studied materials.

Please comment on whether any of the changes particularly in the mechanical properties were great enough to induce failure of the denture.  Were the colour and surface roughness changes acceptable?

Response:  The mechanical properties didn’t change after the selected period of time in such a degree to induce the failure of the denture.

The surface roughness as stated in the discussion section was accepted form the clinical point of view with values for Ra( average roughness ) under 0,2 microns. As well the optical properties were acceptable. 

Please comment on the moisture uptake of PEEK and the potential errors in the weighing, previous studies looking at the use of PEEK as an arthroplasty bearing material have reported extended durations of soaking being required for the material to stabilise eg. Brockett, CL., et al. "Wear of ceramic‐on‐carbon fiber‐reinforced poly‐ether ether ketone hip replacements." Journal of Biomedical Materials Research Part B: Applied Biomaterials 100.6 (2012): 1459-1465. and Cowie, RM., et al. "Wear and Friction of UHMWPE-on-PEEK OPTIMA™." Journal of the Mechanical Behavior of Biomedical Materials 89 (2019): 65-71.  How do the measurements taken in this study compare to previous investigations where rigorous cleaning, drying and weighing protocols have been used?

Response: The protocols in this research followed other research articles in literature that has the same subject. We followed and respected the available protocols, such as ISO 11405 for thermal aging. The samples were weighed on an analytical balance AS 220.R2 PLUS (Sinergy Lab line-RADWAG, Poland) ) with five decimals (0.00001) after each period.

For the nanohardness measurements, please comment on how the values obtained in this study compare to conventional denture materials eg. CoCr.  Would a lower hardness result in scratching of he material?  Would this lead to material loss?  What influence would this have on the aesthetics and cleaning of the denture?  Scratching and third body wear of PEEK has been investigated for orthopaedic implant materials.  PEEK has been shown to scratch easily but when articulating against polyethylene has been shown to have a self-polishing effect (Cowie, R M., et al. "Third body wear of UHMWPE-on-PEEK-OPTIMA™." Materials 13.6 (2020): 1264.)

Response: The microhardness of the samples was smaller compared to other dental materials such as Co-Cr. The imposed conditions simulated the oral environment.

Line 447 – please change ‘extra slight’ to ‘extremely slight’ as per the table

Response: We agree to this point and changed.

Line 482 – please elaborate on the ‘modified’ PEEK.  What materials were used in the modification?

Response: We agree to this point. We added more information.

There are differences between the two 100% PEEK materials.  Do the findings from this study translate to all PEEK materials?  Are the differences due to the batch or manufacturer?

Response: The findings in this study related to the materials included in this study. The producers for both of the non-reinforced PEEK do not elaborated on the composition. The two PEEK materials have different manufacturer’s.

How do the findings from this study influence how PEEK is used in dentistry?  Eg. Should colour matching and manufacture be carried out on pre-soaked material?

Response: The study aimed to see in which degree the immersion and thermal aging affects their properties.  The immersion reproduces the process that takes place in the oral cavity once the restorations are finalised. We wanted to see in which degree the color changes after this process. The difference that take place are not semnificative.

What is the intended processing route for the materials when used in dentistry?  Will the manufacturing process influence the crystallinity of the material particularly on the surface as the rate of cooling of PEEK has been shown to influence its crystallinity and therefore colour.

Response: The studied materials are indicated to be CAD/CAM milled.

Conclusions

Please remove the first sentence of the conclusion (line 493-494)

Response: We removed the first sentence.

Line 505 – please change ‘extra slight’ to ‘extremely slight’ as per the table

Response: We agree to this point and changed the term.

Additional typos

Line 28 – Change ‘Studied’ to ‘The studied’?

Line 30 – change ‘induce’ to ‘induces’

Line 42 – Change ‘strong’ to ‘strongly’

Line 121 – change ‘weight’ to ‘weighed’

Line 452 – change ‘nanomechanic’ to ‘nanomechanical’

Line 489 – change ‘was’ to ‘were’

Line 496 – Change ‘Studied’ to ‘The studied’?

Response: We agree to this point and corrected.

Round 2

Reviewer 2 Report

Dear authors,

Thanks you for your efforts to revise the manuscript. While the overall quality of the manuscript has improved, the ISO you followed (11405) is irrelevant which gives guidance on the testing of the adhesive bond between restorative dental materials and tooth structure. In my opinion, it may not be an adequate standard for testing dental polymer-based restorative materials such as PEEK, including their water sorption, color stability etc. Therefore, the major concern in my original review report remains unchanged. 

Author Response

                Response: Thank you again for the review. We standardised the research respecting the specific information found in research dental literature and producers indications, despite the fact, that there are few studies and information regarding dental PEEK materials.

1.Muhsin SA, Hatton PV, Johnson A, Sereno N, Wood DJ. Determination of Polyetheretherketone (PEEK) mechanical properties as a denture material. Saudi Dent J. 2019 Jul;31(3):382-391. doi: 10.1016/j.sdentj.2019.03.005. Epub 2019 Mar 13. PMID: 31337944; PMCID: PMC6626261.

  1. Shrivastava SP, Dable R, Raj APN, Mutneja P, Srivastava SB, Haque M. Comparison of Mechanical Properties of PEEK and PMMA: An In Vitro Study. J Contemp Dent Pract. 2021 Feb 1;22(2):179-183. PMID: 34257179.
  2. Sunarso, Tsuchiya A, Toita R, Tsuru K, Ishikawa K. Enhanced Osseointegration Capability of Poly(ether ether ketone) via Combined Phosphate and Calcium Surface-Functionalization. Int J Mol Sci. 2019 Dec 27;21(1):198. doi: 10.3390/ijms21010198. PMID: 31892154; PMCID: PMC6981423.

                PEEK samples were milled with a cutting device that has microns precision, in order to obtain equal samples with 15x10x1 mm3. These precise dimensions were selected from the indications of PEEK materials specified in the producer instructions. In this manuscript we determined in which degree thermal aging changes reflect in colour modification, micro and nano-roughness, micro-hardness. We were able to identify the changes in roughness on a micro level and at a nano- level using AFM equipment.   As well the water saturation was determined using an analytical balance with five decimals. Further studies need to be made in order to continue this research in water absorbtion, but this research limited to this aim.

                The ISO standard 11405 refers only for the thermal aging process. We used this standard for the hydro-thermal aging process in this article.

Round 3

Reviewer 2 Report

The overall quality of the manuscript has improved.